# Late Embryogenesis Abundant Protein–Client Protein Interactions

**DOI:** 10.3390/plants9070814

**Published:** 2020-06-29

**Authors:** Lynnette M. A. Dirk, Caser Ghaafar Abdel, Imran Ahmad, Izabel Costa Silva Neta, Cristiane Carvalho Pereira, Francisco Elder Carlos Bezerra Pereira, Sandra Helena Unêda-Trevisoli, Daniel Guariz Pinheiro, Allan Bruce Downie

**Affiliations:** 1Department of Horticulture, University of Kentucky Seed Biology Program, Plant Science Building, 1405 Veterans Drive, University of Kentucky, Lexington, KY 40546-0312, USA; ldirk@uky.edu; 2Agriculture College, Al-Muthanna University, Samawah, Al-Muthanna 66001, Iraq; abdelcaser@gmail.com; 3Department of Horticulture, Faculty of Crop Production Sciences, The University of Agriculture, Peshawar, Khyber Pakhtunkhwa 25120, Pakistan; imran73pk@gmail.com; 4Agroceres, Inc., Patos de Minas, Minas Gerais CEP: 38703-240, Brazil; izabel.neta@agroceres.com; 5Departamento de Agricultura—Setor de Sementes, Federal University of Lavras, Lavras, Minas Gerais CEP: 37200-000, Brazil; cristianecpe@gmail.com; 6Germisul Ltd., Campo Grande, Mato Grosso do Sul CEP: 79108-011, Brazil; eldercarlos12@gmail.com; 7Department of Vegetable Production, (UNESP) National University of São Paulo, Jaboticabal, São Paulo CEP: 14884-900, Brazil; shu.trevisoli@unesp.br; 8Department of Biology, Faculty of Philosophy, Science and Letters of Ribeirão Preto, University of São Paulo, Ribeirão Preto, São Paulo CEP: 14040-901, Brazil; dgpinheiro@gmail.com

**Keywords:** late embryogenesis abundant, protein interaction, desiccation, seed, natural protection and repair mechanism, stress

## Abstract

The intrinsically disordered proteins belonging to the LATE EMBRYOGENESIS ABUNDANT protein (LEAP) family have been ascribed a protective function over an array of intracellular components. We focus on how LEAPs may protect a stress-susceptible proteome. These examples include instances of LEAPs providing a shield molecule function, possibly by instigating liquid-liquid phase separations. Some LEAPs bind directly to their client proteins, exerting a holdase-type chaperonin function. Finally, instances of LEAP–client protein interactions have been documented, where the LEAP modulates (interferes with) the function of the client protein, acting as a surreptitious rheostat of cellular homeostasis. From the examples identified to date, it is apparent that client protein modulation also serves to mitigate stress. While some LEAPs can physically bind and protect client proteins, some apparently bind to assist the degradation of the client proteins with which they associate. Documented instances of LEAP–client protein binding, even in the absence of stress, brings to the fore the necessity of identifying how the LEAPs are degraded post-stress to render them innocuous, a first step in understanding how the cell regulates their abundance.

## 1. Introduction

### 1.1. LATE EMBRYOGENESIS ABUNDANT PROTEINs and Nonreducing Oligosaccharide Accumulation

#### 1.1.1. The Entire Protective Program

The transition from aquatic to terrestrial life overcame significant obstacles, one of which was survival in drier surroundings. This capacity, taken to its extreme, gave rise to desiccation tolerance. Constitutive “extremophile” organisms have adapted to withstand desiccation during any part of their life cycle (e.g., tardigrades, some bryophytes, and resurrection plants). For most anhydrobiotes (“conditional extremophiles”), the capacity to withstand the stresses involved in desiccation is closely associated with a single stage of their life cycle, and their cells (and thus the organism comprised of them) subsequently lose the capacity to survive this severe stress. The majority of organisms succumb to desiccation, regardless of the stage of their life cycle at which this stress occurs. The attributes that can be ascribed to the capacity to withstand desiccation have been extensively examined [1]. The difference between constitutive and conditional desiccation tolerance was first attributed to the possibility that the constitutive extremophiles may have an entire proteome capable of withstanding desiccation. All proteins of constitutive extremophiles could dry and rehydrate with no loss of integrity or function. Such a proteome would require no expenditure of energy by the cell to allow the proteome to survive water loss. Conditional extremophiles would have to manufacture protective molecules for otherwise susceptible proteins and/or translate different paralogous proteins that could withstand desiccation. This capacity is similar to organisms that have adapted a whole proteome to survive, for example, high temperatures (e.g., *Thermus aquaticus*; [2]). However, the advent of whole genome sequencing has refuted this hypothesis in most instances [3]. Specifically, genome comparisons of either constitutive or conditional extremophiles with their closest desiccation-sensitive species has revealed that either form of extremophile is capable of deploying a discrete set of molecular adaptations comprised of protective molecules and morphological alterations that the desiccation-sensitive species cannot [4]. This has been interpreted to mean that cells can withstand desiccation if and when the entire protective program (a “desiccation footprint”; [5]) is implemented. In the constitutive anhydrobiotes, the entire protective program is deployed (or rapidly deployable) throughout the life cycle, while in the conditional anhydrobiote, the entire protective program is implemented possibly only during a discrete phase of the lifecycle. Susceptible species are missing a (or several) crucial elements of the entire protective program and therefore remain susceptible to desiccation throughout their lives [6,7,8]. Even desiccation-tolerant species can be killed [9,10] or destabilized with regard to their longevity in the dry state if part of the entire protective program is removed [11,12].

#### 1.1.2. What Benefits Might Be Realized by Understanding the Entire Protective Program?

The cellular constituents comprising the entire protective program can vary among species [13]. The cells that do not execute the entire protective program at the time of water loss are unable to withstand desiccation, even if they have done so at some other anhydrobiotic stage of their development. The logical conclusion to this thesis is that an understanding of what constitutes the entire protective program, and the capacity to deploy this program to other cells currently without it, will confer upon these cells desiccation tolerance. Furthermore, an understanding of the components of the entire protective program should permit the introduction of these protective molecules to fragile biological materials that would allow the fragile biological materials to be desiccated, stored, and subsequently rehydrated without loss of function. At some point, our greater understanding of how fragile biological materials can withstand extreme desiccation with the aid of molecules that exert a demonstrated (yet poorly understood) protective role will permit fragile biological materials (e.g., drugs, plasma, pharmaceuticals, etc.) to be desiccated, carried, and stored at drastically reduced weight, bulk, and cost. Based on the paradigms observed in nature [14,15], in the desiccated state, the preserved fragile biological materials will also have a greatly enhanced storage resilience and an extended lifespan. When required, the desiccated fragile biological materials can be reconstituted with sterile water, ready for use. Currently, however, the capacity to desiccate fragile biological materials, including macromolecules (enzymes, lipid vesicles, antibodies, sera), therapeutics (vaccines, venom, drug formulations), cells (blood), organs, and even whole organisms, while retaining their biological activity or viability, continues to be a challenge [16]. Nevertheless, the logistical and humanitarian benefits derived from successfully drying fragile biological materials are immense [17,18]. Success in this endeavor would facilitate the ease of fragile biological materials’ transport and storage, avoid the necessity of maintaining fragile biological materials at low temperatures (4 °C and lower), reducing waste and toxicosis through cold-chain failure, while potentiating universal access to these compounds. Robotic aerial delivery of fragile biological materials across difficult dangerous terrain becomes increasingly more feasible the lighter the load of the amalgam (fragile biological materials and the packaging necessary to maintain the fragile biological materials in an active state).

What are the components of the entire protective program deployed to permit cells to survive desiccation? This list is extensive and, as previously mentioned, varies among organisms [13], including not only molecules that exist inside the drying cell but structural alterations to the cell itself. An excellent overview of what constitutes the current knowledge of the entire protective program has been elucidated [1,19]. Included in the requirements for desiccation tolerance, but insufficient to impart it alone (with possible exceptions; [20]), are the accumulation of non-reducing sugars, specifically trehalose, sucrose, and the raffinose family of oligosaccharides, as well as intrinsically disordered proteins. Non-reducing sugars and intrinsically disordered proteins are both positively correlated with desiccation tolerance [20]. Both occur in the cells of most (but not all; [13]) organisms capable of anhydrobiosis, and their accumulation in these organisms is tightly linked with the entry of cells into situations of low free water availability [21]. Of the intrinsically disordered proteins associated with desiccation tolerance, the production of the LATE EMBRYOGENESIS ABUNDANT PROTEINs (LEAPs) group has been positively and tightly correlated with the prelude to entry into a desiccation-tolerant state [22,23,24,25]. The various sugars and the intrinsically disordered proteins are all thought to impart numerous benefits to drying cells, including: 1) The replacement of water, serving to maintain the complex molecular structure of lipids and proteins [26,27]; 2) synergistic generation and stabilization of the amorphous glass within the cells (vitrification) [28,29,30]; 3) LEAPs’ reduction of reactive oxygen species concentrations [31,32]; 4) LEAPs’ action as “molecular shields” preventing protein aggregation through steric interference due to their high concentration within stressed cells [3]; and 5) some LEAPs having the capacity to sequester cations [33]. During desiccation, this cation sequestration capacity is thought to prevent aggregation through calcium concentration regulation, [34] and ion sequestration can mitigate the generation of reactive oxygen species leading to cellular damage [35,36,37]. LEAP ion binding also permits the phloem-mediated transfer of micronutrients (Fe3+) within the plant body [33]. Both water replacement [38] and vitrification [39] are thought to be necessary but insufficient for desiccation tolerance [40] and can be considered complementary protective components of the entire protective program.

### 1.2. Identification of the Role of Non-Reducing Sugars and LEAPs in Desiccation Tolerance

Problems exist when trying to analyze the functional mechanism, at the molecular level, of the non-reducing sugars and/or LEAPs. Foremost among these is that the non-reducing sugars/LEAPs are presumed to exert their protective function in systems where water is scarce, precluding assays in hydrated environments. This has led to numerous examples of chemically altering the solution properties in which these components of the entire protective program are situated to mimic dehydration [41], crowding [42,43], or redox balance [44]. While these methods allow some analysis of LEAP/non-reducing sugars’ behavior, they are still just proxies for the amalgam of stresses with which the desiccated cell is confronted. Problems also include a lack of catalytic properties or “active sites” per se to use in LEAP “assays”. Additionally, LEAPs as intrinsically disordered proteins make tertiary structures conducive to crystallization for X-ray structural analysis impossible to generate [45,46], though potentially feasible and physiologically relevant if bound to their client protein. Furthermore, the non-reducing sugars can exert a protective function when included, at increasingly greater concentrations, either alone (trehalose) or in specific ratios (sucrose:raffinose), to many systems, enzymes, and membranes [12,47]. The non-reducing sugars have a structure hypothesized to assist in water replacement [26]; however, certain non-reducing sugars seemingly must occur in specific ratios to improve efficacy [12,48,49]. Why this is so is currently not understood. There are reports that non-reducing sugars’ protective function increases synergistically when they occur in mixtures with intrinsically disordered proteins, including some LEAPs [20,21,50]. There are a plethora of studies demonstrating increased tolerance to a range of abiotic stresses when either the capacity to produce copious amounts of non-reducing sugars or LEAPs is engineered through overexpression of the relevant genes. These studies can inform us that these protective molecules are able to guard against environmental insult, yet they do not provide information on how this protection is provided. Furthermore, both forms of molecule can occur simultaneously in dehydrating organisms capable of withstanding stress, both share the same intracellular locations, and both, together, must be considered a closer approximation to the entire protective program thought to be necessary to survive anhydrobiosis for the species employing them. In some animal systems, it appears that the use of intrinsically disordered proteins to withstand desiccation is linked to the capacity to also produce the non-reducing sugar trehalose, although this is not universally true as there are animals capable of desiccation that use intrinsically disordered proteins without using non-reducing sugars [13]. In orthodox seeds, LEAPs and some form of non-reducing sugars (or mixture thereof) are part of the natural protection and repair mechanism active during late embryogenesis and after hydration during germination. However, generating organisms with the capacity to accumulate both non-reducing sugars and LEAPs does not provide information on how the combination of these protective molecules imparts protection, if they physically associate with fragile biological material’s, how tightly they do so, or their stoichiometry. It does not identify the fragile biological materials that are in need of protection, precluding a systems biology approach to make more resilient versions of these fragile biological materials.

Referring back to the previously stated premise that deployment of the entire protective program should impart desiccation tolerance to cells/organisms without this capacity, it is thought provoking to understand that there is no single entire protective program used to impart desiccation tolerance among anhydrobiotic organisms but that the components of the entire protective program can vary depending on the species! What does this imply for those hoping to impart desiccation tolerance to organisms/systems without this capacity? Are there several entire protective programs, any of which will be equally successful in providing desiccation tolerance once deployed? Conversely, does this mean that an entire protective program from one set of organisms will not work in a sub-set of different organisms, requiring the successful elucidation and deployment of a specific entire protective program tailored to a class of organisms? While the former case will assist a more rapid understanding of the fundamental requirements for desiccation tolerance that are met by multiple protective molecules, the latter eventuality will make the goal of identifying and deploying an entire protective program more daunting still!

### 1.3. How are the LEAPs Thought to Provide Protection to the Proteome? “Hands off”, “Hands on”, or Both?

#### 1.3.1. Shield Molecule Role

Focusing on organisms that deploy LEAPs during times of stress, there are at least two lines of thought that have evolved regarding how the LEAPs function to safeguard a stress-susceptible proteome. The first posits that they are shield molecules, physically separating cellular entities from each other during desiccation, as crowding within the dehydrating cell becomes more intense or as the opportunity of partially denatured protein aggregate formation increases [50]. This activity does not require that the LEAPs bind to any protein, just sterically hinder interactions among non-LEAPs [3], a “hands off” approach. The second hypothesis extends the first, suggesting that, in addition to a shield molecule role, physical binding of a LEAP and its specific client protein repertoire (“hands on”) also occurs, which contributes to client protein stability [42,51,52]. There can be little doubt concerning the shield molecule function. The very fact that LEAPs, expressed in systems that do not encode them, nevertheless impart certain forms of stress tolerance to them, resulting in generally superior endogenous protein stabilities [21], argues for a shield molecule capacity. Thus, even a single LEAP, faced with a non-conspecific proteome, can positively impact the solubility of most (all?) of these proteins, and positively influence the survivability of the cell during stress [53]. Additionally, there are many studies that have used non-conspecific labile proteins from organisms without an anhydrobiotic stage in their life cycle as reporter enzymes to demonstrate LEAP-protective properties in vitro [54]. A study based on observations made on thawing solutions containing LEAPs revealed how the LEAP preferentially occupies the liquid–air interface, excluding other proteins from this region, forcing them to reside in a more hydrated sub-domain as the liquid containing them freezes [55]. This capacity is one mechanism by which the LEAPs could act as a shield molecule [55] and reveals a property of some LEAPs potentially allowing them to form liquid-liquid phase separations, recruiting client proteins into a droplet in which the client protein is more protected without necessarily invoking a LEAP–client protein physical interaction. The possibility that some LEAPs do form or promote liquid-liquid phase separation in vivo is receiving attention [13]. This shield molecule capacity provides some hope that specific entire protective programs will not be necessary to impart desiccation tolerance to organisms without it. An entire protective program may be universally effective by altering the properties of the solvent rather than the properties of the solute.

#### 1.3.2. Liquid-Liquid Phase Separations. Could LEAPs be Involved?

Although water seems like a simple molecule, there is still fundamental controversy regarding how this liquid sustains life [56]. The properties of the molecule are influenced by the environment in which it exists and, thus, is exceptionally complicated to understand in the context of a cell [56]. Intrinsically disordered protein-induced liquid-liquid phase separation is a mechanism by which the cell creates non-membranous organelles [57]. As previously suggested [58,59], LEAPS, by virtue of their intrinsically disordered nature, may be well suited to assist in the formation of proteinaceous membrane-less organelles via liquid-liquid phase separation, though it may be only the subset that have the required hydrophobic nature that can participate. Liquid-liquid phase separation could permit isolation of certain cellular components from each other as the water is removed from the cell. Such a protective mechanism could be easily reversible as the microenvironment changes, relying on the intrinsic properties of the LEAPs’ amino acid makeup to assist in both the formation and dissolution of the proteinaceous membrane-less organelle to prevent deleterious interactions of various cellular components. The post-translational modification of intrinsically disordered proteins also changes their capacity to generate proteinaceous membrane-less organelles [60]. Not only are intrinsically disordered proteins a population of protein structures, but the post-translational modification of those intrinsically disordered proteins creates a regulatory means of controlling the population’s involvement in liquid-liquid phase separation and sensitivity to specific micro-environments. Many of the LEAPs are subject to post-translational modifications (Appendix A), the relevance of which have largely been unresolved to date, with the exception of phosphorylation of some of the dehydrins [61].

If the LEAPs are forming liquid-liquid phase separation, how are client proteins recruited into these structures and is this a selective process? It is tempting to speculate that if LEAPs are essential to liquid-liquid phase separation, then each of the LEAPs involved in liquid-liquid phase separation are identifying and binding to their specific client proteins and this drives recruitment. However, this would necessitate protein–protein interaction. This would complicate the assignment of the actual protective property exerted by the LEAP in a liquid-liquid phase separation. Is the protection strictly a shield molecule functionality, housing the susceptible proteins in a liquid micro-environment, conducive to the client protein’s stability and binding only to recruit client proteins into the protective shell, or does binding to the client protein also assist in protection? Is it possible that if LEAPs set up a liquid-liquid phase separation, susceptible client proteins diffuse toward and into the more protective shell without ever engaging the LEAPs physically, constituting a legitimate shield molecule function? The addition of other solutes, e.g., specific ratios of sucrose to raffinose family oligosaccharides, may influence the propensity, timing, and types of liquid-liquid phase separation that occur as all molecular concentrations are increasing and mobility of molecules are changing as the glassy state forms during desiccation. Likewise, micro-environment alterations, and presumably liquid-liquid phase separation properties, will vary as, for example, the orthodox seed intracellular environment constantly changes through various cycles of imbibition and drying. Furthermore, as temperature alterations, and other abiotic factors, change during germination, this will also influence liquid-liquid phase separation and the protection they afford [57,58].

A particularly interesting characteristic of purified LEAPs is their ability to change their conformation in response to an alteration in the osmotic potential of the environment [42]. Stress-dependent disorder-to-order transitions of LEAPs or other intrinsically disordered proteins could represent a novel sensing mechanism for the formation of liquid-liquid phase separation able to prevent stress-induced irreversible aggregation of cellular proteins, as recently shown for two yeast intrinsically disordered region-containing proteins [62,63]. Testing whether or not disordered plant LEAPs assist the formation of liquid-liquid phase separation could reveal more information on how plants organize their cells under stressful situations. Greater methodological advancement may be required to investigate the role of such transient cellular structures, particularly during desiccation.

#### 1.3.3. LEAP Overexpression, or Heterologous Expression, and Tolerance to NaCl

Although accounts vary depending on how membership in the LEAP family is classified, there are at least 51 LEAPs widely acknowledged to be encoded in the *Arabidopsis thaliana* genome [46,64]. There are only five instances in the literature where *Arabidopsis* plants overexpressing *Arabidopsis* LEAPs have been tested on high NaCl. Three instances resulted in plants with superior salt tolerance at some stage of the life cycle while two instances showed reduced tolerance to NaCl (Appendix A). The two instances with reduced tolerance were both for overexpressed LEA4 proteins while the superior NaCl tolerance resulted from overexpressing a LEA2, a LEA4, or a SMP family member (Appendix A). The lack of documented studies in which *Arabidopsis* LEAP OE plants were tested for NaCl tolerance (5 out of 51) is a knowledge gap due to either a lack of testing (there are 15 of the *Arabidopsis* LEAPs that have been overexpressed in *Arabidopsis*) or a lack of reporting negative results (LEAP-OEs were not more resistant to NaCl than the empty vector controls). For non-*Arabidopsis* LEAPs ectopically expressed in other organisms, including insect cell lines [65], when tested on NaCl, the consequences have usually been superior NaCl tolerance (Appendix A) relative to the respective negative control. While this may be due to a fixation of testing transgenic organisms overexpressing LEAPs on high salt-containing media because this is what has been done successfully in past studies, there are instances where the clearest results concerning tolerance to a diversity of applied stresses has been for salt stress. The tolerance of organisms ectopically expressing LEAPs to high NaCl transcends any particular LEAP family, with examples in the literature from the LEA2, 3, 4, SMP,, and dehydrin families, and is particularly prevalent for organisms ectopically expressing proteins from the ASR LEAP family (Appendix A). ASR (abscisic acid (ABA), stress, ripening-induced) proteins have been considered as a LEAP family [23]. The N-terminus has a metal binding capacity that both stabilizes the protein against proteases and permits DNA binding. Such a universal positive influence of LEAP’s presence is suggestive of a general shield molecule function capable of orchestrating a universally entire protective program in the face of hyper-saline (NaCl) stress.

#### 1.3.4. LEAP–Client Protein Physical Interaction Conferring Protection: Chaperonin “Holdase” Function

The evidence supporting at least some of the LEAPs physically interacting with proteins, ostensibly to protect them from denaturation, is accumulating and is incontrovertible. LEAP–client protein physical binding offers a solution to the fact that a number of very different agronomic traits have been tightly linked to quantitative trait loci centered on genes encoding specific LEAPS (at least those of the dehydrin family; [66]). Assuming that the LEAPs are causal loci attributed to the observed difference in these diverse phenotypes, the specific client proteins with which the strongest LEAP allele is binding and protecting can result in an observable difference in agronomic traits based on the level of protection afforded to the client proteins. This diversity of phenotypes would be impossible if the same client proteins are protected by each LEAP allele, so the conclusion must be that different LEAPs protect different client proteins, leading to an enhancement of different agronomic traits. LEAP binding to specific client proteins can also explain why upon mutation of a single LEAP in organisms with many co-localized co-expressed members available for complementation nevertheless results in a distinct phenotype (Appendix A). The very fact that in *Arabidopsis*, there are at least seven to nine distinct families of LEAPs [64,67,68] and different plant species contain a or several LEAP families that are not present in other plants (e.g., ASR) implies different LEAP functions, which would not be the case if a single LEAP family, at least one member of which is present in all subcellular compartments, is sufficient to impart stress tolerance as a shield molecule. Certainly, the capacity of some LEAPs to physically bind to membranes, thus altering their properties, is very well established [27,61,69] and the same capacity for specific client proteins is becoming increasingly evident (see below).

Despite early arguments against a physical interaction between LEAPs and client proteins based on the failures of protein pulldown assays to demonstrate physical associations, subsequently, a variety of protein–protein interaction assays, with or without an applied stress, have demonstrated LEAP–client protein physical associations. These assays include yeast two hybrid [70,71], phage display [72], bimolecular fluorescence complementation [73], luciferase complementation [74], and tandem affinity purification [75]. Additionally, the recent development of thermophoretic assays [76], not just quantitatively demonstrates LEAP–client protein binding but also documented a greater affinity in solution between LEAPs and their client protein as the stressor (H_2_O_2_ in this case) increases [44]. LEAP–client protein interactions have been documented among LEA2, dehydrin, LEA4, and SMP LEAP families (Appendix A), providing specific evidence of LEAP–client protein interactions, surprisingly, even among the LEAPs themselves.

The benefit of LEAP–client protein binding has largely been assumed to be the protection of the client proteins from some form of stress; however, for reasons provided above (no catalytic capacity, intrinsically disordered, etc.), it is difficult to determine how LEAP binding imparts protection to a specific client protein. This quandary is very similar to that for microprotein research, where the consequences of microprotein presence are most readily explored by determining to what the microprotein binds [77]. The same approach taken in the case of the LEAPs (yeast two hybrid, phage display, etc.) has revealed specific examples of LEAP–client protein physical interaction and these will be examined below. The documentation of a specific LEAP–client protein physical association allows testable hypotheses to be devised concerning the consequence of the interaction. This has revealed a plethora of mechanisms by which these “protective proteins” exert their function to maintain the homeostasis of the stressed cell through a specific client protein association.

#### 1.3.5. LEAPs: Engaging in Client Protein Protection and/or Acting as Surreptitious Rheostats of Cellular Homeostasis

There are instances where it seems that specific LEAPs impart protection to client proteins, binding to and preserving the client protein from permanent loss of function as and while it is stressed, and disengaging to become innocuous upon rehydration. However, other situations have been documented, where LEAP–client protein binding improves homeostasis through means other than client protein protection. So focused is the community studying LEAP function on the protective role of these intrinsically disordered proteins, we may be blind to additional consequences of LEAP–client protein binding. Is this focus on LEAP–client protein binding necessarily revealing a single LEAP functionality, that of client protein protection? Or is it diverting attention away from auxiliary functions of LEAP–client protein interaction, e.g., the modulation of cellular homeostasis? In fact, have some LEAPs capable of binding client proteins been secunded, in certain instances, to specific protein–protein interactions designed to alter client protein function to mitigate stress in addition to (or instead of) protecting the client protein from stress? Certainly, the fact that LEAPs may encompass multi-functionality has been recognized previously [78]. Recognizing that LEAPs may have a repertoire of client proteins to which they bind allows the possibility that, under a particular stress, a LEAP may protect some client proteins while interfering with the function of others and that this association (or even the identity of the client proteins to which a LEAP can bind) may change as the stress changes.

#### 1.3.6. How to Recognize Client Protein Protection Versus Interference With Client Protein Function?

If LEAPs can alter client protein functionality, then the phenotypes of any single LEAP mutant become more nuanced. If the LEAP is protecting the client protein during stress, then mutating the LEAP should result in a diminution of the client protein function during stress and the phenotype of the *leap* organism under stress would mimic the phenotype of the *client protein* in the same organism under stress, a so-called “parity of phenotypes” (Figure 1a,c). Similarly, overexpressing the LEAP, if protection at endogenous LEAP amounts is sub-optimal, should result in the same phenotype (additional stabilization and hence, influence of the client protein) as overexpression of the client protein itself (Figure 1d). However, if the LEAP is intervening in the localization or engagement of the client protein with other cellular constituents (Figure 1b), then the consequences of mutation or overexpression of either the LEAP or its client protein will be quite different. Now, a parity of phenotypes for opposite mutations/overexpression results (Figure 1e,f). What do documented examples of LEAP–client protein interaction show us?

#### 1.3.7. Known LEAP–Client Protein Interactions

##### LEAP–Client Protein Interactions Where LEAP Binding Protects the Client Protein from Abiotic Stress

A dehydrin from *Arabidopsis thaliana*, ERD14 (At1g76180), interacts with the *Arabidopsis* Phi9 GLUTATHIONE-S-TRANSFERASE9 (At2g30860). Phi9 associates with the dehydrin in ratios of 1:5, (or greater) respectively to acquire protection from H_2_O_2_-related structural damage, presumably to the carboxy-terminus substrate binding site, which is otherwise rendered hyper-flexible by oxidation [76]. ERD14 also binds to CATALASE, protecting this enzyme. In addition to protection, binding of ERD14 may enhance the activity of these two enzymes, which are directed to reducing oxidative stress ([76]; Figure 2). How enzyme activity might be increased upon LEAP binding is not currently understood.

The phosphorylation, or any post-translational modification (PTM) of a LEAP, demonstrates at least transient LEAP binding by a protein-modifying enzyme. However, this is quite different from the LEAP binding to impart protection or to interfere with client protein action. The LEAP-modifying enzyme, like a protease degrading a LEAP, cannot be considered a client protein based strictly on binding (Figure 1). Nevertheless, from a strictly pragmatic view, PTM of the LEAP must be of some consequence, even if it is not currently obvious, and so we have included three examples (depicting only two of them) of LEAP PTM here. All are of LEAPs being phosphorylated by various kinases, the implications of which are currently the matter of speculation.

A histidine-rich dehydrin from *Arabidopsis thaliana* HIRD11 (At1g54410) interacts with the *Arabidopsis* leucine-rich repeat receptor-like kinase (LRR-RLK) PHLOEM INTERCALATED WITH XYLEM-LIKE 1 (AtPXL1; AT1G08590). Plasma membrane-localized AtPXL1 phosphorylates cytoplasmically or plasma membrane-localized HIRD11. The assertion that phosphorylation leads to plasma membrane association of HIRD11 is equivocal (see question mark in Figure 3); based on the comparisons of GFP signal from AtHIRD11-EGFP to the YFP signal with bimolecular fluorescence complementation images of AtHIRD11–YFP^N^ interaction with AtPXL11-YFP^C^, where only the sub-set of AtHIRD11 interacting with the intrinsic membrane protein AtPXL1 is detectable ([79]; Figure 3). If phosphorylation does lead to membrane binding, it is quite different from the dehydrin AtLti30, which, in its phosphorylated state, is released from membrane binding [61]. Although the consequences of LEAP phosphorylation are not currently known and there is no evidence that the LEAP stabilized AtPXL1 in any way, this interaction has been placed under this rubric.

A CALCIUM/CALMODULIN BINDING CYTOPLASMIC RECEPTOR-LIKE KINASE2 (CB-RLK2) positively influences plant tolerance to both hypersaline (NaCl) and alkaline (carbonate) conditions when overexpressed [80,81]. The LEAP GsPM30 binds GsCBRLK2. Overexpression of GsPM30 also enhances plant tolerance to NaCl (Figure 1d; [80]). The authors speculate that the association leads to phosphorylation of GsPM30 (see question mark in Figure 4). Whether GsPM30 protects the cytoplasmically localized receptor-like kinase from NaCl stress is not currently known. The association of CB-RLK2 with a membrane-spanning receptor kinase as depicted (Figure 4) is an assumption we have made based on the general nature of such kinase cascades [82]. Whether the LEAP binds CB-RLK2 to allow it to function in times of stress or whether the kinase phosphorylates the LEAP and in that state the LEAP exerts a protective function unrelated to maintaining CB-RLK2 stability is also unknown.

The SUCROSE NONFERMENTING RECEPTOR-LIKE KINASE2.10 (SnRK2.10), an SnRK2 unresponsive to ABA, binds and phosphorylates both ERD10 and ERD14 LEAPs during osmotic stress. While ERD10 maintains a cytoplasmic distribution whether it is phosphorylated or not, the population of ERD14 subsequently resides in both the cytoplasm and nucleus [83]. The direct relevance of dehydrin phosphorylation for the protection of client proteins or other cellular constituents for either LEAP is currently unknown.

The *Oryza sativa* LEA 2 family gene, OsLEA5 (LOC_Os05g50710), is upregulated by ABA, at least partially due to the action of the rice ZINC FINGER C2H2 PROTEIN (OsZFP36; LOC_Os03g32230; Figure 5), the transcription of which is itself induced by both ABA and H_2_O_2_. As well, when OsLEA5 is present, OsZFP36 protein is more stable. This enhanced stability results in prolonged OsZFP36 DNA residency (and increased *OsLEA5* expression) in a positive feedback loop [84,85,86]. Additionally, increasing OsZPF36 transcriptional influence due to OsLEA5 protection may account for many of the reported physiological alterations and differences in ABA sensitivity and redox poise that enhance tolerance to oxidative stress. For example, an additional target gene of OsZFP36 is *Oryza sativa ASCORBATE PEROXIDASE 1* (*OsAPX1*; BAA08264; [87]; LOC_Os03g17690), the protein from which mitigates the accumulation of H_2_O_2_, one form of damaging reactive oxygen species, and thus assists oxidative stress tolerance (Figure 5).

A dehydrin LEAP from the cactus *Opuntia streptacantha*, when overexpressed in *Arabidopsis*, led to enhanced cold tolerance [88]. This dehydrin as well as three from *Arabidopsis* (COR47, ERD10, and RAB18; see Appendix A for details for abbreviations) were found to bind to the aquaporin AtPIP2B (At2g37170) on the inside face of the plasma membrane [89] (Figure 6). Because both AtPIP2B and the dehydrins influence cold tolerance positively when overexpressed (a parity of phenotypes (Figure 1d)), these authors speculate that there may be a functional relationship in the LEAP–client protein interaction because aquaporins are known to be susceptible to denaturation that can be minimized in another species (at least during heat stress) by association with a chaperonin (ALPHA-CRYSTALLIN) [90]. If we assume that the mechanism of aquaporin denaturation due to supra- or sub-optimal temperature perturbations is the same as in planta, AtPIP2B association with the cactus dehydrin, or any of the three con-specific dehydrins, may protect AtPIP2B from cold stress denaturation in the same manner as ALPHA-CRYSTALLIN during heat stress [89]. However, the dehydrins bind only to the inside face of the aquaporin/plasma membrane (Figure 6) while the aquaporin denatured by heat stress converted predominantly alpha-helical conformations to predominantly beta-sheets when denatured [90]. The most prominent alpha-helices in functional aquaporins are those forming the hydrophobic transmembrane domains, buried within the plasma membrane. Of the three LEAPs binding the aquaporin (and each other), ERD10 had been previously identified as a plasma membrane-associated protein that accumulates in response to cold acclimation [91]. Using mass spectrometry, the DHNs ERD10 and ERD14 were also identified as accumulating in the membrane fraction of cold-acclimated plants but may be easily detached, suggesting that these proteins are only peripherally associated with the plasma membrane in response to cold acclimation [92]. So, it would seem that only a small portion of the aquaporin is accessible to the LEAPs and that portion only on the cytoplasmic side of the membrane.

How the influence of the dehydrins is exerted to maintain lipid association with the hydrophobic trans-membrane portions of PIP2B or by somehow locking the PIP2B into a stable structure in the membrane remains to be elucidated. Regardless, the overexpression of the dehydrins or their client protein aquaporin [88,93] enhanced cold tolerance, demonstrating a parity of phenotypes for the overexpression of either (Figure 1d). Overexpression of the aquaporin itself, resulting in enhanced cold tolerance, may be due strictly to stochastic variation in denaturation within the available population (and/or over time) of aquaporins. Increasing PIP2B numbers ensures that there is a sufficiently large sub-set of the population that evades denaturation at cold temperatures in the overexpressors to maintain a threshold water transfer capacity within the membrane. Overexpression of any of the LEAPs stabilizes PIP2B, which also ensures a threshold water transfer capacity. In addition, the same *Arabidopsis* dehydrins (COR47, ERD10, and RAB18) were found to form homo- and hetero-dimers within and among themselves, although the functional significance of these various associations has yet to be determined [73] (Figure 6).

Loss-of-function mutants in the SEED MATURATION PROTEIN1 (At3g12960), the closest *Arabidopsis* ortholog of the soybean (*Glycine max*) LEAP, GmPM28 (NM_001251057; GLYMA08G172800; [94]), somehow influenced the hydrated seed memory of supra-optimal temperature stress experienced during seed germination. Unlike Wild Type seeds, *smp1* seeds lose the capacity to enter thermo-dormancy when removed to permissive temperatures [95]. The SMP1 and GmPM28 orthologs were able to bind several of the same client proteins [72]. The client protein to which both orthologous LEAPs bound most frequently, regardless of supra-optimal temperature stress, was the CANCER SUSCEPTIBILITY CANDIDATE3 (CASC3) also known as METASTATIC LYMPH NODE 51 (MLN51), a core member of the exon junction complex (EJC) thought to participate in several post-transcriptional mRNA processes, including intracellular transport [96], translation enhancement [97], and intron-based nonsense-mediated mRNA decay [98]. CASC3 has been described as a “poorly folded protein” [99], implying that it is structurally unstable. The region of the client proteins (including CASC3) bound by the LEAP orthologues was the same, consisting of the client protein site that was the most hydrophilic or among the most hydrophilic of the entire client protein (Figure 7; [100]). The implications of this commonality among client protein sites bound by the two LEAPs may be nothing more than that the hydrophilic regions of the client proteins would be most likely to be solvent exposed and available for LEAP binding. Concerning the mechanism behind the protection of the seed hydration memory by the LEAP, one possibility is that CASC3 is present on mRNA from a heat-responsive gene, encoding a protein repressive to the completion of germination. At supraoptimal temperatures, CASC3 may be destabilized unless bound by SMP1. Upon the return to a permissive temperature, without CASC3 boosting the translation of mRNA formed in response to (during) stress, insufficient inhibitor is produced to prevent the completion of germination. While the veracity of this hypothesis has yet to be tested, many LEAPs have been positively associated with stress memory in *Arabidopsis* [101] while another mutation in a seed maturation protein family member (At3g22490) also leads to a reduction of dormancy [102], and so this is the scenario that has been depicted (Figure 7).

Using immunoprecipitation to compare proteins binding to a chloroplast stroma-localized LEAP from cold-acclimated and naïve plants, [103] found that COLD-REGULATED 15A (COR15Am; m refers to mature chloroplast-localized LEAP) bound to both the large and small subunits of RIBULOSE-1,5-BISPHOSPHATE CARBOXYLASE/OXYGENASE (RUBISCO) when plants had been acclimated to the cold but not prior to this. Regardless of cold acclimation, COR15am was able to form homo-oligomers (usually tetramers) in vivo [103]. They speculated that the COR15am–RUBISCO association prevents freezing-induced deactivation of RUBISCO (complex disassembly, partial unwinding, or lessened aggregation; [104]), which could partially account for the enhanced RUBISCO activity upon cold acclimation (Figure 8; [103]). COR15 has been shown to stabilize enzymes previously [105] and its overexpression had a demonstrable positive influence in freezing stress tolerance [106]. When both *COR15A* and *COR15B* were knocked down by RNA interference (RNAi), a reduction in freezing tolerance, regardless of acclimation, resulted [107]. When RUBISCO activity was assessed for freezing tolerance in RNAi lines and Wild Type that had both been acclimated, surprisingly, the RNAi lines had increased RUBISCO freezing tolerance [107]! Equally confusing, while COR15A bound to membranes of the plastid, the reduced freezing tolerance registered in acclimated cor15a/cor15b RNAi plants was associated with membrane disruption of the plasma membrane (assessed by ion leakage) relative to the plastid (assessed by chlorophyll fluorescence) [107].

At least two dehydrin proteins (At1g20440, COR47; At1g20450, ERD10) were found to interact with actin. ERD10 was investigated further and found to stabilize actin polymers while inhibiting both the initiation of actin growth and its depolymerization (Figure 9; [108]). In the presence of latrunculin B (constituting an artificial stress), the initiation of growth, although delayed, was not prevented, as it was in the absence of the LEAP, suggesting that ERD10 could bind to both the polymer and to individual actin units prior to their addition to the plus end (Figure 9; [108]).

By far the best example of the multifarious nature of a LEAP associating with multiple client proteins and exerting either a protective function for some while interfering with the client protein’s function/stability for others, the consequences of which, nonetheless, all lead to a plant more adapted to drought stress, comes from a series of papers from the Dong/Wang collaboration. A Medicago truncatula cold stress-upregulated LEAP of the dehydrin family (Medicago truncatula COLD-ACCLIMATION SPECIFIC PROTEIN31; MtCAS31; Medtr6g084640) was found to also accumulate during other abiotic stresses, including dehydration stress [109]. The client proteins with which MtCAS31 interacts, and the consequences of this interaction, underlines the complex nature of how one LEAP can orchestrate stress tolerance, once stress has upregulated the LEAP’s expression. The MtCAS31 is highly similar to the *Arabidopsis* ERD10 (At1g20450) and ERD14 (At1g76180) dehydrins. *MtCAS31* is upregulated by drought stress and, when CAS31 is present, it binds one of several leghemoglobin proteins (MtLb120-1; Medtr5g080440) and protects MtLb120-1 against denaturation during supra-optimal temperatures and drought stress. The continued functioning of leghemoglobin during stress, safeguarded by MtCAS31, maintains the appropriate oxygen level in nodules conducive to both nitrogenase activity and bacterial respiration, supporting nodule persistence allowing the legume to retain functional nodules and nitrogen fixation capacities under drought conditions [110]. Here, the knockout phenotype of *mtcas31* suggested the protection of leghemoglobin by the dehydrin ([110]; Figure 1a,c and Figure 10).

##### LEAP:Client Protein Interactions Where LEAP Binding Interferes with Client Protein Function

When tested in yeast two hybrid assays against an *Arabidopsis thaliana* prey library, MtCAS31 was found to also bind INDUCER OF CBF EXPRESSION1 (ICE1) (also known as SCREAM1 (SCRM1) [109,111]) and the Medicago ICE1 orthologous protein, Medtr7g083900. When *MtCAS31* was overexpressed in *Arabidopsis*, the association of MtCAS31 with SCRM1 downregulated stomatal numbers in developing leaves, presumably by sequestering SCRM1 away from its association with other basic HELIX-LOOP-HELIX transcription factors. Because the basic HELIX-LOOP-HELIX heterodimers were not formed, the complex could not bind their cognate promoters and cell fate was altered away from the meristemoid lineage [112]. The consequences of this reduction in stomatal numbers was a decrease in transpirational water loss and greater drought tolerance [109] (Figure 11). Based on published accounts [111], *SCRM1* overexpression results in phenotypes opposite of those for *MtCAS31* overexpression (Figure 1e,f), pointing to the capacity of the dehydrin to interfere with the function of the basic HELIX-LOOP-HELIX leucine zipper transcription factor.

The Dong/Wang collaboration has also shown that MtCAS31, when present during drought stress, binds to an aquaporin, MtPIP2;7 (Medtr2g094270). The LEAP also binds to MtAUTOPHAGY 8F (MtATG8F; Medtr4g037225) through two distinct ATG8-interacting motifs located near the amino terminus of the dehydrin. During drought stress, MtCAS31 binds the aquaporin and introduces it as cargo to the autophagosome through its simultaneous interaction with MtATG8F. Both the dehydrin and the aquaporin are degraded in this manner, reducing the water channel abundance in the plasma membrane of the roots such that hydrolytic conductivity between the plant and the soil is reduced ([113]; Figure 12). Apparently, the role of dehydrin binding to aquaporins varies depending on the stress imposed because, during cold stress [89], dehydrin binding to AtPIP2B stabilized the water channel in the membrane to improve cold tolerance (Figure 6).

##### Instances Where LEAP–Client Protein Binding Results in Interference or Protection of the Same Client Protein, Depending on the Abiotic Stress

Using yeast two hybrid, Li et al. [70] found that the AtLEA1 protein (At1g01470; AtLEA14 in NCBI) binds to (or is bound by) the E3 ubiquitin ligase F-Box protein, ARABIDOPSIS PHLOEM PROTEIN 2-B11 (AtPP2-B11; At1g80110), a protein that accumulates during drought. Both *atpp2-b11* RNAi lines and *LEA1*-overexpressing lines have enhanced ABA sensitivity. This similarity in ABA sensitivity can be explained by the observation that the AtPP2-B11 F-BOX protein targets the SUCROSE NON-FERMENTING KINASEs (SnRK2.2, 2.3), reducing the titer of these kinases in the cell. These SnRKs are positive regulators of ABA signal transduction, so when they are present, ABA sensitivity is increased. For obvious reasons, reducing F-BOX titer through RNAi stabilizes the kinase titer and results in increased ABA sensitivity (Figure 13a). The observation that LEA1 binds and stabilizes the AtPP2-B11 protein may mean that the LEA1 sequesters the F-BOX protein away from its target proteins, such as SnRK2.2 and 2.3, also increasing ABA sensitivity by indirectly preserving the SnRK2s from polyubiquitination (Figure 13a). There is no evidence that the “stabilization” of ATPP2-B11 by LEA1 results in the F-BOX being degraded if left in the SCF complex without cargo or that the LEA1 acts as an alternative F-BOX substrate by performing a suicidal substitution for the kinases and saturating the F-BOX and thus indirectly stabilizing the kinases. A later study determined that the LEAP binds the FBOX protein, stabilizing the FBOX protein during salt stress [114]. However, during salt stress, overexpressing the FBOX protein results in salt tolerance, which is the same phenotype observed when the LEAP is overexpressed (Figure 13b). This is consistent with the conclusion that the LEAP binds and protects the FBOX (at least during salt stress), leading to greater salt tolerance [114]. In the instance of salt stress tolerance, the similarity between the LEAP OE and the FBOX OE is an example of a parity of phenotypes resulting from the same ectopic expressions (Figure 1d and Figure 13b). The results from the two studies (one on drought and one on salt tolerance) can be reconciled by considering the possibility that the LEAP binds the FBOX to prevent it capturing its substrate(s) and/or from docking with the *Arabidopsis* SKP-LIKE (ASK) proteins in E3 ligase complexes during drought stress but that the LEAP binds the FBOX to protect it during salt stress, permitting the FBOX to target a different group of proteins that would otherwise decrease salt tolerance. This conclusion requires that the FBOX have multiple target proteins (or the same target protein) that can increase (drought) or decrease (salt) tolerance (Figure 13). The authors considered the possibility that the FBOX targets the LEAP for polyubiquitination during salt stress, leaving the LEAP client proteins without protection, but they examined the protein stability of the LEAP during FBOX upregulation and determined LEAP stability was uninfluenced [114].

### 1.4. Following the Completion of Germination or Recovery from Stress Leads to LEAP Declines: How Are They Degraded and How Is This Controlled?

The LEAPs, with few exceptions, are upregulated due to stress, and the proteins from the LEAPs accumulate to assist as part of the entire protective program of many desiccation-tolerant organisms. The newly accumulating LEAPs then acquire their various client proteins either by an association resulting from binding-induced alterations prompted by their client proteins, or the best adapted LEAP configuration is selected by the client protein [57]. So, LEAP–client protein interactions form when stress initiates LEAP transcription and translation and may be further incited (affinity increased) due to stress-induced alterations in the client protein, the LEAP, both, or by alterations in the intracellular environment, which has deviated considerably from “normal”. We know that LEAPs can bind their client proteins even in the absence of stress [72]. This capacity should be detrimental to plant growth in the absence of stress when LEAPs are overexpressed. Yet, there have only been four reports of a yield penalty in LEAP-overexpressing organisms (see below). Potentially then, LEAP presence is insufficient to physically engage their client proteins with high affinity, loose association being adequate for client protein binding in screens, such as yeast two hybrid or phage display, but inconsequential, in vivo, to influence client protein affinity for its(their) target(s). This has been demonstrated for LEAP membrane associations, where dehydrins do not commence folding in the presence of membranes until dehydration initiates crowding [115]. Similarly, LEAP–client protein affinity may increase as the intracellular environment changes due to stress, but simply increasing LEAP amounts through overexpression is insufficient to instigate noticeable phenotypes. Similarly, as the intracellular milieu reacquires its unstressed properties, the LEAPs are anticipated to disengage their client proteins, which are stable without the LEAPs. However, even if this scenario is true for all LEAP–client protein associations, a substantial amount of cellular nitrogen and carbon is tied up in the highly abundant LEAPs during stress or orthodox seed development, which is further exacerbated by overexpression.

The LEAPs associated with desiccation tolerance or stress resistance accumulate to considerable abundance in the cell, one reason for their discovery in cotton embryos: They were hard to miss [116,117]! In desiccating tissue, the abundance of the LEAPs brings them into contact with other cellular constituents, including other intrinsically disordered LEAPs. This is particularly the case when LEAPs are overexpressed, further boosting the endogenous LEAP compliment, or when ectopically expressed in organisms that may or may not have an endogenous repertoire of LEAPs also available for deployment. Why are these interactions not “seeding” protein aggregation rather than being responsible for the opposite [118,119]? In other instances of persistence/accumulation of proteins containing substantial intrinsically disordered regions, they have induced departures from homeostasis through the promotion of toxic accumulations of the intrinsically disordered proteins in interactions with each other or in non-self-associations leading to disease (e.g., Huntingtin, tau, etc.; [45,120]). There are other possible associations in which the LEAPs may play a role (e.g., proteinaceous membrane-less organelles/ liquid-liquid phase separation), the stability and timing of formation of which may lead to pathology [57,121]. The expanded polyQ tracts associated with the severity and time of onset of, for example, Huntington’s disease, are disordered, polar, and soluble regions [122,123] that, when they persist, cause endoplasmic reticular stress [122] among other dysfunctions. Why is this not the case for any of the families of LEAPs yet tested (Appendix A)? One possibility is that most of the manifestations of disorders resulting from intrinsically disordered proteins’ hyper-accumulation in animals appear to influence the cells of the nervous system (e.g., Alzheimer’s, Parkinson’s, and Huntington’s disease). Studies on LEAP overexpression have occurred in plants, bacteria, and yeast, lacking a nervous system, or in non-neuronal animal cell types [21,124,125]; however, this perceived link between intrinsically disordered protein-mediated pathology and neuronal cells is probably inaccurate. When human kidney cells were used with a transiently expressed Huntington’s disease first exon extended to a polyQ tract of 76, which was capable of forming aggregates within this non-neuronal tissue, co-expression of an inducible LEAP in the same cells did not exacerbate aggregation but rather served to ameliorate the aggregation of the Huntington’s disease protein [119]. There must be something fundamentally different between these intrinsically disordered proteins.

The vast majority of the LEAP overexpression studies have reported enhanced fitness during stress of the organisms expressing them (Appendix A). For those studies that have had antibody to the LEAP under study, the protein abundance increases upon overexpression (e.g., LEA4-5; [51]) and tends to persist (e.g., [126]). Yet, in certain instances, overexpression seems to be balanced by an upregulated LEAP degradation capacity or a post-transcriptional block to translation such that only stress induction results in hyper-accumulation of the LEAP (e.g., compare COR15A-OE or COR15B-OE amounts without and with cold acclimation [107]). Despite the advantage the LEAPs confer upon tissue during stress, their persistent hyper-accumulation upon over/ectopic expression cannot be viewed as innocuous in non-stressed tissues. This is particularly true if the LEAPs are capable of altering interactions to modify cellular physiology in the absence of stress (see above). Their persistence in the non-stressed cell may be disruptive to homeostasis and their abundance certainly deprives the cell of amino acids that could be used to synthesize more relevant proteins [127] redirected towards repair and growth in the new unstressed situation. Nevertheless, we could find only a few reports of a yield penalty suffered upon LEAP ectopic expression. The determination of yield penalties for transgenic plants, particularly those subjected to tissue culture, is fraught with difficulties, described in detail in [128], and the observed penalties potentially have nothing to do with the ectopic expression of the transgene per se. These authors point out in their study of rice overexpressing *OsLEA3-1* that, regardless of the promoter used in the construct (2X*CaMV35S*, *ACTIN1*, or LEAP (*HVA-like*)), under normal conditions, the first generation usually had significantly reduced fertility and grain yield, relative to Wild Type. Once specific lines were chosen containing a single transgene insertion and made homozygous, the yield penalties under normal growth were eliminated while yield under drought conditions remained superior to Wild Type [128].

With this caveat in mind, the *Arabidopsis* LEAPs *COR15A* and *COR15B*, when overexpressed, both delayed flowering and resulted in a smaller rosette size relative to Wild Type [107]. While flowering time determines neither seed numbers nor seed weight (yield) as long as the plants eventually flower to the same degree as Wild Type [129], the photosynthate available for mobilization for seed production is correlated with leaf area. We assume therefore that the smaller rosettes caused a yield penalty. Potatoes overexpressing either a dehydrin or a LEA3 under the control of a constitutive (CaMV35S) or stress-induced (COR78) promoter performed less well than untransformed Wild Type when grown under field conditions in years when rainfall was not severely limited [130]. Although southern blots were performed and yield tested over multiple years, it is unclear if the lines had single insertions or if the lines were homozygous. However, the yield penalty was greater for CaMV35S, relative to the stress-inducible COR78 promoter [130], so we are assuming there is a yield penalty. Additionally, using the CaMV35S promoter to drive a tomato *ASR1* gene in transgenic potato, tuber numbers were significantly reduced [131]. No information on homozygousity or transgene numbers were provided. However, the harvest index of the OE lines was uninfluenced relative to Wild Type, prompting the authors to state that while tuber numbers were lower, the overall tuber size was greater than Wild Type [131], although these data were not shown. So, in this instance, no yield penalty results. Reduced seed numbers per plant without influencing seed weight have been reported for two maize lines homozygous constitutively expressing the maize LEAP *RAB28* using a *UBIQUITIN* promoter. Single insertions for these lines were established from segregation ratios in back crosses to Wild Type [132]. A report specifying the necessity of using a stress-inducible promoter to avoid a yield penalty in transgenic mulberry is available [133]. In total, and to the best of our ability to ascertain, these four instances constitute the reports of yield penalties observed for plants overexpressing LEAPs.

By tracking LEAP persistence following the completion of germination or the alleviation of stress in normal tissues, these proteins are clearly decimated by the cells after the completion of germination [51,134], upon hatching [135], or following the alleviation of stress [136,137]. In fact, the LEAPs’ decline in abundance after the stressful situation has passed is part of the spectrum of circumstantial evidence interpreted to mean that they have a stress-protective function [138] and post-stress, lose their relevance for homeostasis, or their loss during prolonged stress leads to deleterious consequences [139]. Yet, their degradation appears to be rather specific, occurring in the cell while many other proteins are stable. While the decline in the LEAPs is well documented, there are few reports on how such a specific reduction occurs.

One means to rapidly draw down the LEAP concentration when the proteins are no longer required is to prevent transcription and destroy the mRNA encoding them, at least ensuring that no further augmentation of the LEAP reserve is possible and assuming that the LEAPs have a high turnover rate. The *LEAP* transcript abundance, following the completion of germination, is rapidly reduced [140], although the pace at which they are degraded may vary, even among members of the same family [51], and the same is true of the cell post-stress [127]. There are multiple indications that the *LEAP* genes are under epigenetic control, transcription from them being rapidly shut down, with rare exceptions [141], following the fundamental switch from late embryogenesis to germination [142], orchestrated by maturation desiccation [143,144,145,146,147]. While many of the *LEAPs* are epigenetically silenced in the seed to seedling transition, some *LEAPs* are still available for transcription throughout the vegetative stages and so are not controlled epigenetically and remain available to assist in vegetative stress tolerance. Maintaining *LEAPs* available for transcription throughout the life cycle rather than epigenetically silencing them characterizes differences between the constitutive and conditional extremophiles mentioned in the introduction.

Repressing *LEAP* transcription might be coupled with an active mechanism to degrade stored mRNA following imbibition. There is a report of a microRNA in tea (*Camellia sinensis*) targeting a *LEAP* transcript [148] and another targeting a *LEAP* of *Arabidopsis* [149], but miRNA targeting *LEAP* mRNA does not appear to be a general phenomenon. Other than these two examples, the assignment of microRNAs to their targets has not identified a single *LEAP* transcript target [150] nor even an miRNA targeting a member of the larger group of protective proteins [151,152]. The mRNA of at least three *Arabidopsis LEAPs* is bound by RNA binding proteins, with consequences for transcript accumulation in the plant [153,154], but this does not appear to degrade the mRNA.

Evidence that the LEAP proteins persist longer than the mRNA encoding them exists [51,155], but disconnects between transcript abundance and protein presence have been documented [156], which have led to suggestions that considerable post-transcriptional regulation determines translation for some LEAPs [134]. In one extreme instance, LEAP residence in the cell, following the alleviation of stress, was considerably longer than most other proteins associated with protection against stress. This is the case for the dehydrin LTI29, which was slow to decline after a week of low temperature followed by a 2-day recovery at normal temperatures following cold stress [157]. The authors suggest that the reduction of most of the proteins associated with cold stress tolerance enables the plant to re-prioritize growth and development while preserving the LEAP as a hedge against recurring cold stress.

A greater percentage of the intrinsically disordered proteins from yeast (*Saccharomyces cerevisiae*) were found to contain a (PEST) domain [158] than the proteins comprising the ordered proteome [120]. The PEST domain (protein sequence rich in these four amino acids “PEST”) was proposed as a potential signal to proteases [158,159]. While there is a report of a PEST domain contributing to the rapid turnover of a dehydrin [160], a survey of the *Arabidopsis* LEAP assemblage revealed few (12 in 8 LEAPs; Appendix A) high-confident sites using the ePESTfind server (http://emboss.bioinformatics.nl/cgi-bin/emboss/epestfind). Intriguingly, though only computer predictions, 7 of the 12 potential PESTS were present in 4 out of 10 *Arabidopsis* dehydrins and these four were all members of the acidic dehydrins while none of the four neutral/basic dehydrins have predicted PEST domains (Appendix A). An examination of dehydrins across species [160] reported that approximately 38% of dehydrins examined contained PEST domains though the number of LEAPs including predicted PEST sites were insufficient to explain the loss of these proteins upon completion of germination or stress alleviation. This raises the interesting possibility that certain families of LEAPs may have different means of degradation. Nevertheless, for three of the *Arabidopsis* LEAPs in root extracts, the half-lives of these proteins were calculated for the non-PEST containing LEA2, At2g44060, as 28 h while the two PEST-containing dehydrins, At1g20450 and At1g76180, had half lives of 52 and 30 h, respectively [161]. These half-lives are not curtailed to the point where the PEST sequences appear to be functionally meaningful, at least not in roots or under the conditions analyzed. There is a report of a dehydrin acting as an adaptor protein in selective autophagy where the dehydrin is also degraded (Figure 12).

While there are many metabolic consequences attributed to protein phosphorylation, one of prominence is to act as a signal for F-Box recognition, leading to polyubiquitination and proteolytic degradation [162]. In total, 26 of the 51 *Arabidopsis* LEAPs are known to be differentially phosphorylated (Appendix A; [163]; Appendix A). However, there are documented examples of how phosphorylation is a mechanism by which LEAPs (at least some dehydrins) alter their affinity for binding cellular components [61]. In addition, there are two of the nine families of LEAPs in *Arabidopsis* for which no phosphorylation has been identified to date (AtM and LEA3; Appendix A). Some instances of interactions of LEAPs with components of the poly-ubiquitination complex have been documented. However, in these instances, there is more evidence to suggest that the LEAP is stabilizing (or interfering with) the component of the polyubiquitination machinery with which it is interacting with rather than being targeted for degradation. As discussed previously, regarding AtLEA1, the details regarding degradation via the F-BOX with which it binds were equivocal (Figure 1 and Figure 13). At this time, there is no compelling evidence to suggest that the LEAPs are decimated using the 26S proteasomal pathway.

One study of metacaspases in two-day-old seedlings found that some of the LEAPs still present at this stage were targets of these rather novel proteases [164]. The seven LEAPs identified as METACASPASE9 (At5g04200) substrates all belong to the LEA4 family. One well-studied initial series of cleavages of a LEAP, initiating its decline, is with the wheat (*Triticum aestivum*) Em protein. A cysteine endopeptidase (to which metacaspases belong; [165]) cuts the LEAP in two sites and, subsequently, the Em protein is completely degraded within 24 h of imbibition [166,167]. None of the metacaspases produce considerable transcript in mature seeds but several are upregulated in the hypocotyl and particularly the radicle, immediately following the completion of germination (e-FP Browser; [168]). The metacaspase endoproteases (if involved in LEAP degradation) only initiate LEAP decline and other peptidases must be responsible for completing degradation.

## 2. Conclusions

The LEAPs have been determined to afford protection to cellular constituents during stressful situations. This review has focused on how LEAPs may protect a stress-susceptible proteome. Hypotheses of how this protection is provided to client proteins include a shield molecule functionality that may or may not prove to be associated with a presumed capacity for LEAPs to orchestrate liquid-liquid phase separation within the stressed cell. The shield molecule functionality does not require LEAP:client protein physical association, although recruitment of client proteins into liquid-liquid phase separation, if indeed orchestrated by LEAPs, may prove to require transient LEAP–client protein associations. Additional means by which LEAPs protect proteins include physically binding to the client protein, a chaperonin functionality that does not require the expenditure of energy, similar to the “holdases” of the heat shock protein family. Examples of such physical association leading to protection have been provided and discussed as have examples where LEAP–client protein affinity increases as the applied stress increases. This requirement of a stress-driven increase in LEAP–client protein affinity seems to explain why there is no general yield penalty for LEAP overexpression. Finally, LEAP–client protein associations may result in the modulation of client protein operation, which has unveiled a third mechanism by which LEAPs tweak the intracellular environment to maintain homeostasis in the face of stress. The identification of the client protein repertoire of individual LEAPs is imperative if the subtleties of the full range of the protective mechanism embodied in these “enigmatic” proteins is to be appreciated. Moreover, identifying the client protein repertoire of a LEAP using methods that inherently permit the identification of where in each client protein a LEAP is binding allows the client protein regions to be scrutinized for commonalities that may inform the molecular topology of the final interaction, allowing more resilient client proteins to be engineered [100]. There is a paucity of information on how the LEAPs are degraded to allow amino acid recovery and recycling following stress alleviation, a necessary means by which the LEAPs are rendered innocuous.

## Figures and Tables

**Figure 1 plants-09-00814-f001:**
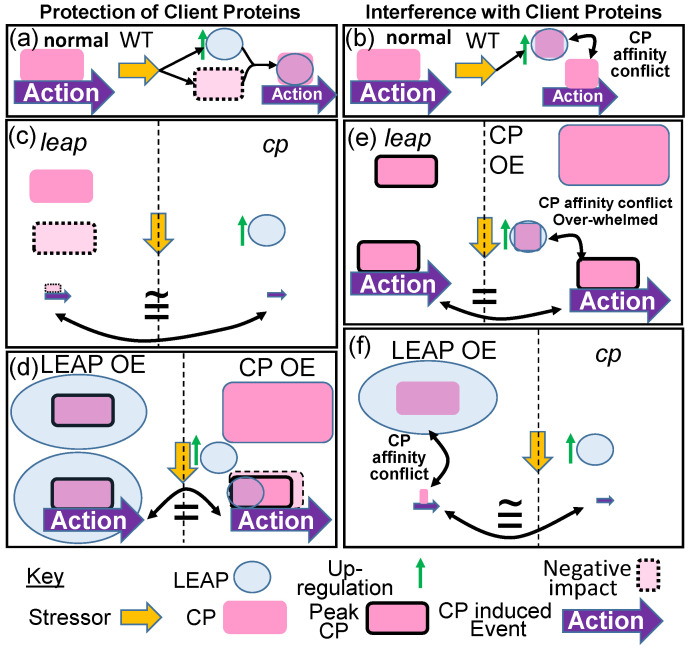
If one accepts that some LEAPs have an affinity for specific client proteins (CP), physically binding to them, the consequence of this is traditionally thought to be client protein protection. (**a**) In Wild Type (WT) organisms, deviations from unstressed (normal) conditions by some stressor upregulates LEAP expression, increasing LEAP protein, and the LEAP chaperonin protects the client protein that would otherwise be decimated by the stress. The client protein continues exerting at least some of its action. (**b**) One alternative to this chaperonin role for LEAPs is that of a surreptitious rheostat of cellular homeostasis. Upon upregulation by the stressor, LEAPs’ affinity for the client protein competes with the client protein affinity for one of its targets, sequestering sufficient client protein away from its target to consequentially mitigate its action. Biotechnological alteration of the LEAP and client protein amounts can provide insights into the nature of the LEAP–client protein relationship. (**c**) If the LEAP protects the client protein from a stressor, a parity of phenotypes results for mutation of either the *leap* or the *client protein* (*cp*). In one instance, the client protein is unprotected, decimated by the stress, and the action is attenuated. In the other, there is no *client protein* and the action is diminished to a basal response. (**d**) Overexpression of either the LEAP (LEAP OE) or the client protein (CP OE) results in either endogenous client protein amounts that are maximally protected by superabundant LEAP or a superabundance of client protein with endogenous LEAP protection, respectively. Either scenario results in a substantially greater portion of the client protein surviving the stress, relative to WT (compare Action arrow in (**d**) eliciting peak client protein action). (**e**,**f**) The client protein binds to some cellular entity and prevents/initiates an action leading to a phenotype. Here, we assume initiation of an action. The LEAP binds some portion of the client protein population, interfering with the initiation of the action (conflict). (**e**) A parity of phenotypes results from the *leap* mutant, allowing the entire client protein population to initiate action or the client protein overexpressor that overwhelms the LEAP binding capacity and can still initiate peak action. (**f**) The elimination of the *client protein* minimizes the action to its basal response, which, depending on the affinity of the LEAP for the client protein, is similar to LEAP overexpression, which sequesters the entire endogenous client protein population, preventing the initiation of the action. Maximal client protein amounts, initiating peak action, are designated by a solid black border. Dashed lines separate the genotypes eliciting the same phenotype.

**Figure 2 plants-09-00814-f002:**
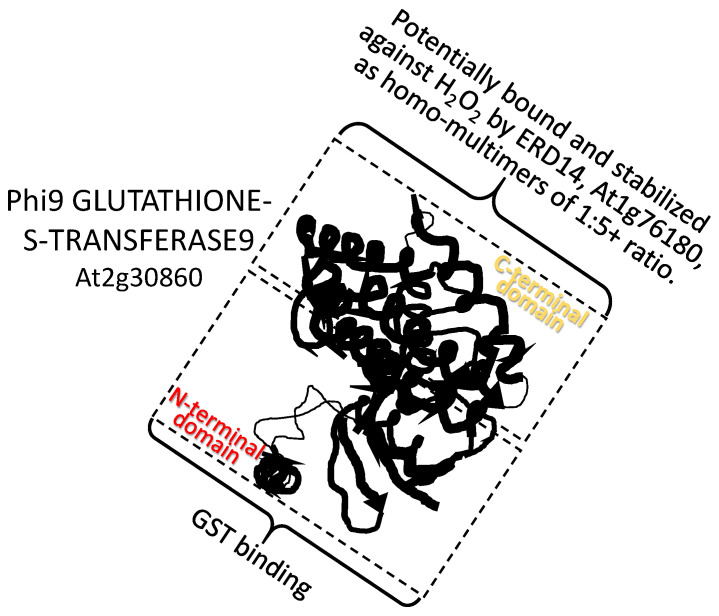
A dehydrin from *Arabidopsis thaliana* EARLY RESPONSE TO DEHYDRATION 14 (ERD14) (At1g76180; pfam dehydrin) interacts with the C-terminal domain of the *Arabidopsis* Phi9 GLUTATHIONE-S-TRANSFERASE9 (At2g30860). Phi9 associates with the dehydrin in ratios of 1:5, respectively (or greater), to acquire protection from H_2_O_2_-related structural damage. GST binding: the glutathione-binding site (G-site).

**Figure 3 plants-09-00814-f003:**
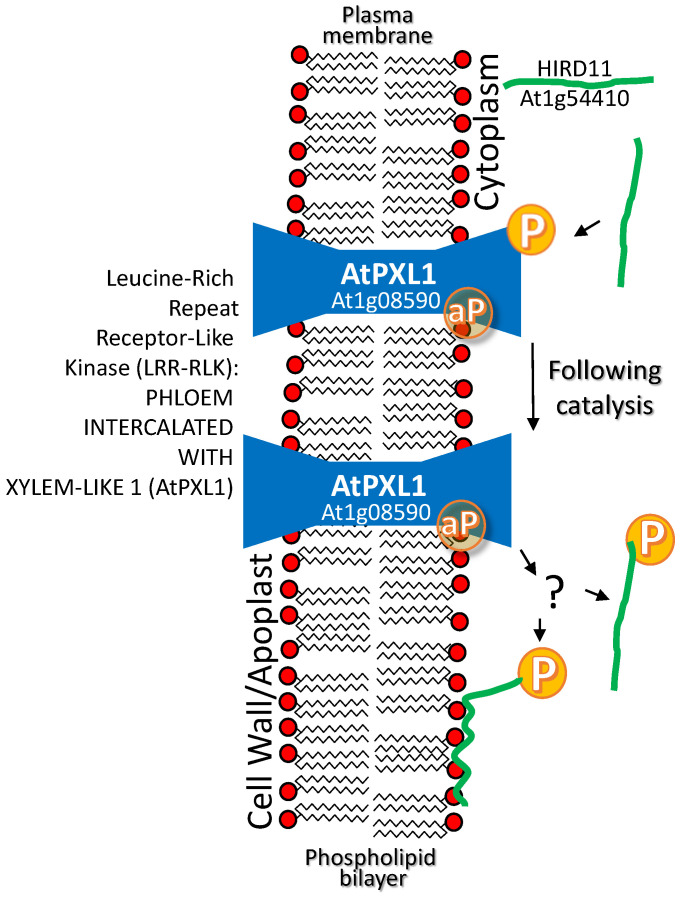
A histidine-rich dehydrin from *Arabidopsis thaliana* HISTIDINE-RICH DEHYDRIN OF 11 KDA (HIRD11) (At1g54410) interacts with the *Arabidopsis* leucine-rich repeat receptor-like kinase (LRR-RLK) PHLOEM INTERCALATED WITH XYLEM-LIKE 1 (AtPXL1; AT1G08590). Plasma membrane-localized AtPXL1 phosphorylates (P) cytoplasmically localized (or plasma membrane associated?) HIRD11. At least, a sub-population of HIRD11 associates with the plasma membrane or AtPXL1 on the plasma membrane. The aP in a yellow circle represents the autocatalyzed phosphate group that is present on most LRR-RLK proteins, though this remains to be defined for this particular enzyme as present or as the phosphate group that gets transferred to the substrate.

**Figure 4 plants-09-00814-f004:**
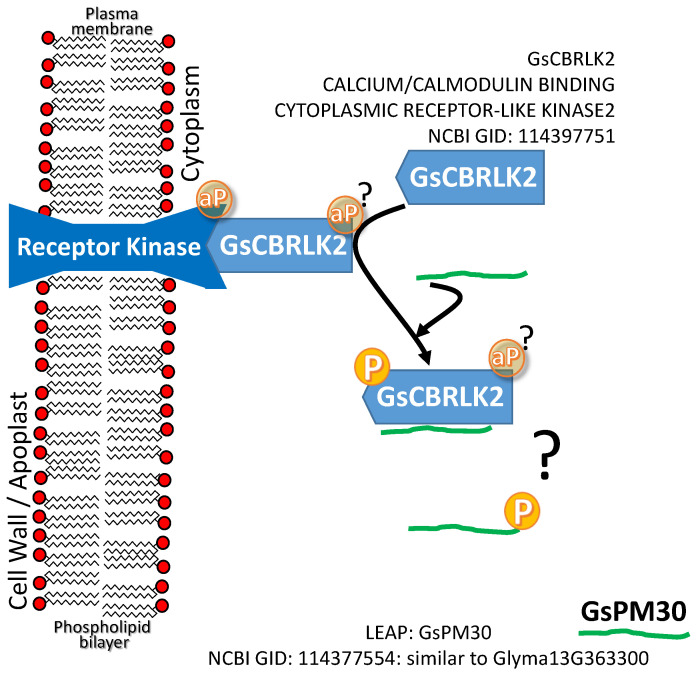
A CALCIUM/CALMODULIN BINDING CYTOPLASMIC RECEPTOR-LIKE KINASE2 (CB-RLK2) and the LEAP. GLYCINE SOJA PHYSIOLOGICAL MATURE30 (GsPM30) that binds GsCBRLK2; both positively influence plant tolerance to hypersaline (NaCl) conditions when overexpressed. It is not currently known if the association leads to phosphorylation (P) of GsPM30 by CB-RLK2 (see question mark above). The aP in a yellow circle represents the autocatalyzed phosphate group that may be present on both the receptor kinases, though this remains to be defined for these particular enzymes as present or as the phosphate group that gets transferred to the substrate.

**Figure 5 plants-09-00814-f005:**
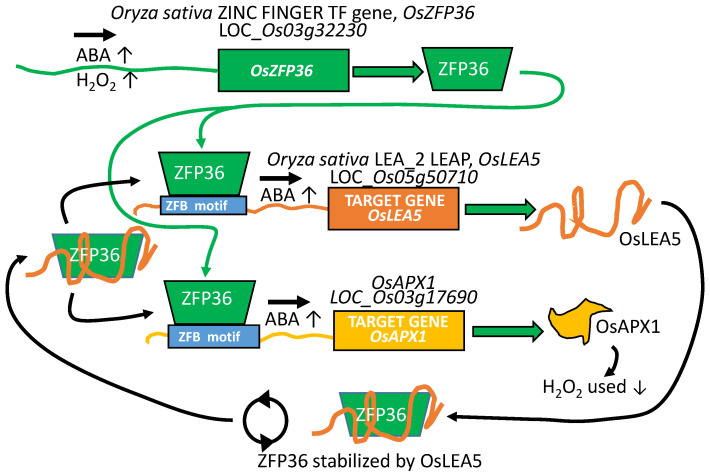
The *Oryza sativa* gene, OsZFP36, a zinc-finger transcription factor, is upregulated by the shock hormone abscisic acid (ABA) and by the reactive oxygen species H_2_O_2_. OsLEA5 (LOC_Os05g50710; pfam LEA 2) is upregulated by ABA, at least partially due to the action of the rice ZINC FINGER C2H2 PROTEIN OsZFP36 (LOC_Os03g32230). When OsLEA5 is present, it stabilizes OsZFP36. This results in enhanced OsZFP36 residency and increased OsLEA5 expression, a positive feedback loop (circular arrows). Another target upregulated by OsZFP36 is the peroxidase *OsAPX1*. This enzyme detoxifies H_2_O_2_, a reactive oxygen species that upregulates *OsZFP36* expression. By recognizing the stabilization of the transcription factor during stress by the protein product of its *LEAP* target, many of the reported physiological alterations and differences in ABA sensitivity and redox poise in common between mutants in the *oslea5* or in *oszfp36* can be explained. ZBF motif: zinc finger binding motif.

**Figure 6 plants-09-00814-f006:**
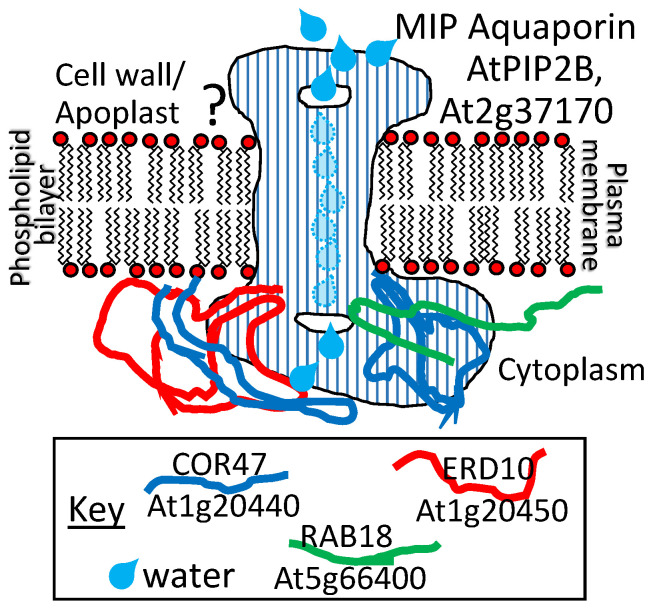
A dehydrin from the cactus *Opuntia streptacantha* (Genbank ID: HM581971) or any of three homologs from *Arabidopsis*
*thaliana* COLD-REGULATED 47 (COR47), EARLY RESPONSIVE TO DEHYDRATION 10 (ERD10), and RESPONSIVE TO ABA 18 (RAB18) (dehydrins), encoded by *At1g20440*; *At1g20450*, and *At5g66400*, respectively, interact with the *Arabidopsis* membrane intrinsic protein (MIP) aquaporin ARABIDOPSIS THALIANA PLAMAMEMBRANE INTRINSIC PROTEIN2B (AtPIP2B), encoded by *At2g37170*. The subcellular distribution of both the LEAPs and this client protein along the plasma membrane is consistent with biochemical evidence of these interactions. The functional significance of this interaction has not yet been elucidated. The three different dehydrins can also associate with each other in homo- or hetero-dimers. The implications of these interactions are also unknown.

**Figure 7 plants-09-00814-f007:**
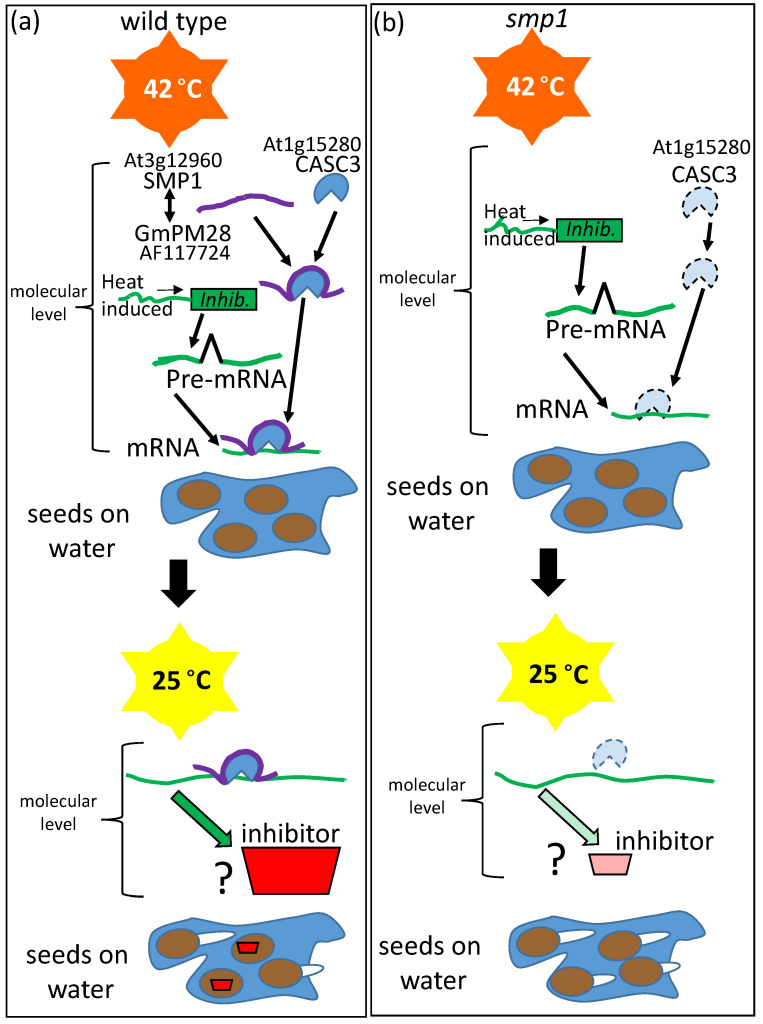
Unlike (**a**) Wild Type *Arabidopsis* seeds, (**b**) mutant *seed maturation protein1* (*smp1*) seeds lose the capacity to enter thermo-dormancy (invoked by 4 days at 42 °C while hydrated (blue pool of water)) when subsequently removed to a permissive temperature, 25 °C. The soybean LEAP GLYCINE MAX PHYSIOLOGICAL MATURE 28 (GmPM28) and its *Arabidopsis* orthologue, SMP1, were both able to bind several of the same client proteins. The protein most frequently bound by both orthologs was the CANCER SUSCEPTIBILITY CANDIDATE3 (CASC3), capable of marking the splice junction on mature mRNA and enhancing translation. Presumably, loss of the LEAP destabilized CASC3 during high-temperature stress and this may lead to the inability of a (presumably repressive) protein to subsequently exert thermo-dormancy on a sub-population of the seeds.

**Figure 8 plants-09-00814-f008:**
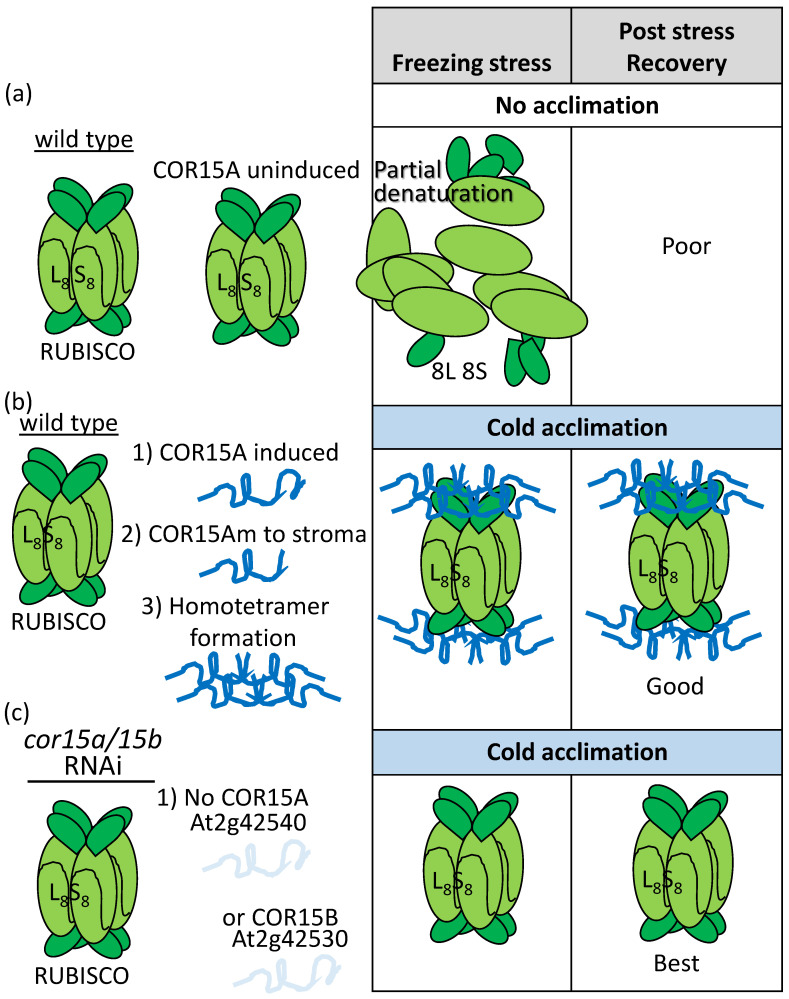
Immunoprecipitation of proteins physically associating with a chloroplast-stroma-localized LEAP, COR15Am (At2g42540), consistently retrieved both the small and the large subunits of RIBULOSE-1,5-BISPHOSPHATE CARBOXYLASE/OXIDASE (RUBISCO) from cold-acclimated (**b**,**c**) but not (**a**) unacclimated plants. This association (**b**) has been positively correlated with a greater capacity to tolerate freezing, a capacity that is enhanced in both protoplasts and chloroplasts overexpressing COR15Am (Appendix A). However, (**c**) RNAi lines reducing COR15A and COR15B simultaneously did not suffer a greater loss of RUBISCO activity during freezing after cold acclimation but rather the opposite was true. L_8_: Eight large RUBISCO subunits; S_8_: Eight small RUBISCO subunits; 8L and 8S are these subunits disassembled from the holoenzyme.

**Figure 9 plants-09-00814-f009:**
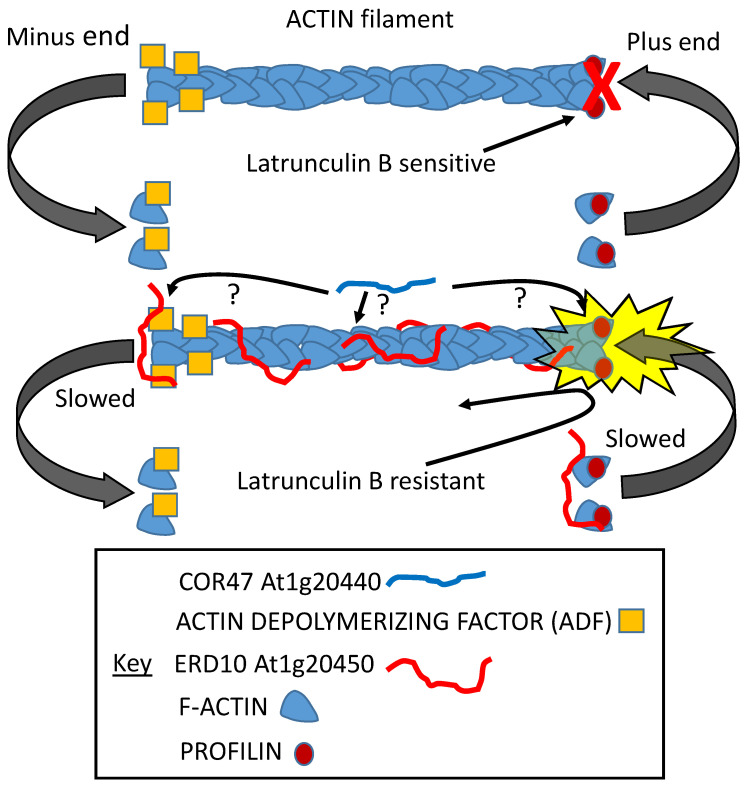
Both COR47 and ERD10 dehydrins were shown to bind ACTIN filaments. A series of experiments demonstrated that ERD10 was probably capable of binding the plus and minus ends as well as the body of the filament. Although COR47 was also shown to bind the filament, where it binds is still unknown. ERD10 binding inhibits nucleation, slowing the initiation of growth while it also slows the depolymerization of the filament. It can interfere with the capacity of latrunculin B to prevent polymerization [108].

**Figure 10 plants-09-00814-f010:**
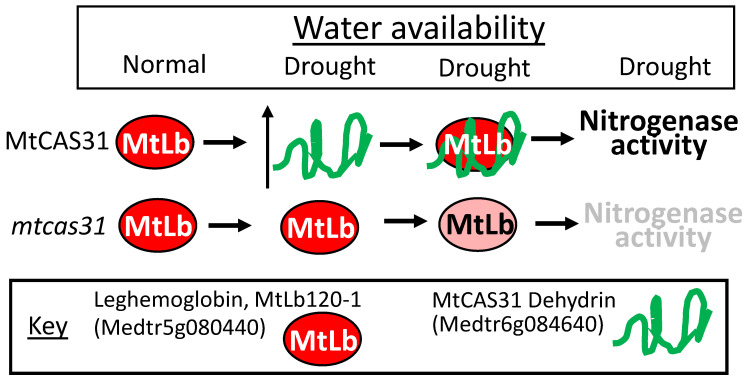
The *Medicago truncatula* dehydrin gene, *MtCAS31* (*Medtr6g084640*; highly similar to the *Arabidopsis ERD10* (*At1g20450*) and *ERD14* (*At1g76180*) pfam dehydrin) is upregulated by drought stress. When MtCAS31 protein is present, it binds one of several leghemoglobin proteins (MtLb120-1; *Medtr5g080440*), protecting it from dehydration induced denaturation. The continued functioning of leghemoglobin, safeguarded by MtCAS31, maintains the appropriate oxygen level in nodules conducive to both nitrogenase activity and bacterial respiration, supporting nodule persistence.

**Figure 11 plants-09-00814-f011:**
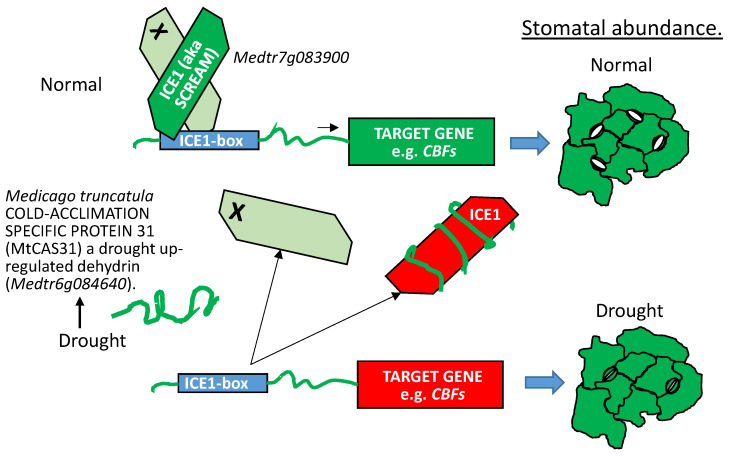
*MtCAS31* (described in Figure 10) is upregulated by drought stress. When MtCAS31 dehydrin protein is present, it binds INDUCER OF CBF EXPRESSION (ICE1, i.e., SCREAM) as well as the Medicago basic HELIX-LOOP-HELIX -LEUCINE ZIPPER transcription factor most homologous to ICE1, Medtr7g083900. The molecular consequences of MtCAS31:ICE1 binding is that the number of ICE1 proteins available to hetero-dimerize with any of the basic HELIX-LOOP-HELIX transcription factors involved in meristemoid initiation, progression to guard mother cells, or the formation of two guard cells (in *Arabidopsis*: SPEECHLESS, MUTE, or FAMA), decreases. The phenotypic consequence for leaves formed when MtCAS31 is present is fewer stomata per unit leaf area, resulting in a physiology tuned to reduce evapotranspiration. X = orthologs of *Arabidopsis* basic HELIX-LOOP-HELIX transcription factors, SPEECHLESS, MUTE, and FAMA but not the basic HELIX-LOOP-HELIX -LEUCINE ZIPPER proteins ICE1 (i.e., SCREAM) and SCREAM2. bHLH-LZ: basic HELIX-LOOP-HELIX-LEUCINE ZIPPER transcription factor; *CBFs*: *C-REPEAT BINDING FACTORs*. Green TF: Binding with its partner basic HELIX-LOOP-HELIX transcription factor to cognate DNA motifs. Target gene(s) are transcriptionally active (green). Red TF: Bound by MtCAS31 and sequestered away from its basic HELIX-LOOP-HELIX partner and DNA motifs. Target gene(s) are transcriptionally repressed (red).

**Figure 12 plants-09-00814-f012:**
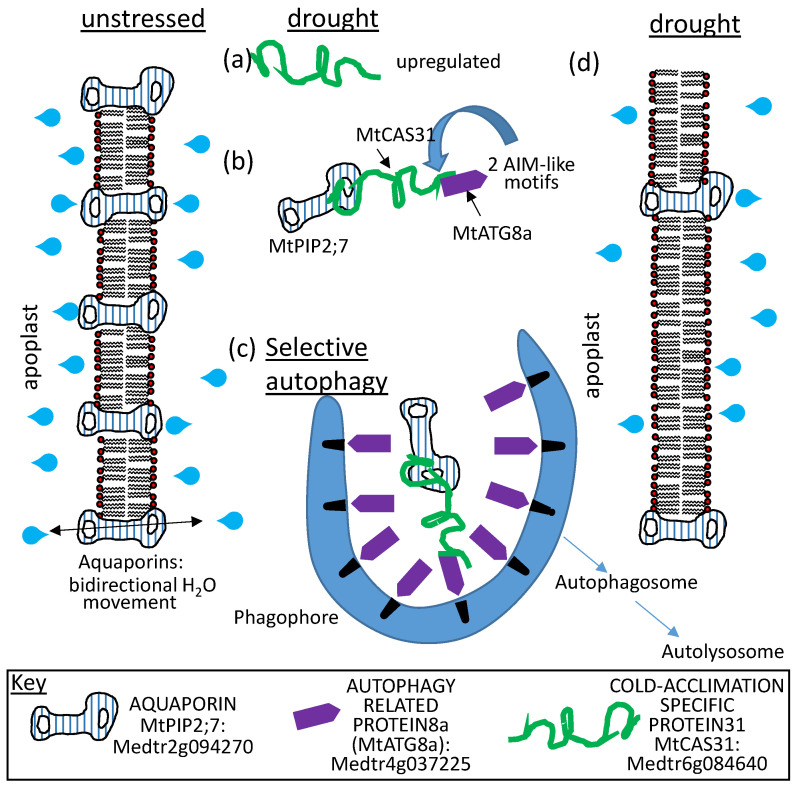
MtCAS31 (described in Figure 10) is: (**a**) upregulated upon dehydration and the protein it encodes; (**b**) binds the plasmamembrane intrinsic protein aquaporin MtPIP2;7 (*Medtr2g094270*) as well as the Medicago AUTOPHAGY-RELATED GENE8 Protein (ATG8a; *Medtr4g037225*) through the 2 AIM-like motifs (Autophagy Interacting Motifs) located near the MtCAS31 amino terminus. Thus, MtCAS31 acts as an adaptor protein; (**c**) directly linking its cargo protein MtPIP2;7 with ATG8a in phagophores, which leads to the destruction of the aquaporins (and MtCAS31) through selective autophagy. While stress is present, so too is MtCAS31, which ensures a continuous turnover of MtPIP2;7. The molecular consequences are a reduction in the number of aquaporins located in the plasma membrane of drought-stressed *Medicago* cells. The phenotypic consequences for the cell when MtCAS31 is present are: (**d**) an altered hydraulic conductivity of the cell membrane, resulting in a physiology tuned to reduce water transport across the cell membrane.

**Figure 13 plants-09-00814-f013:**
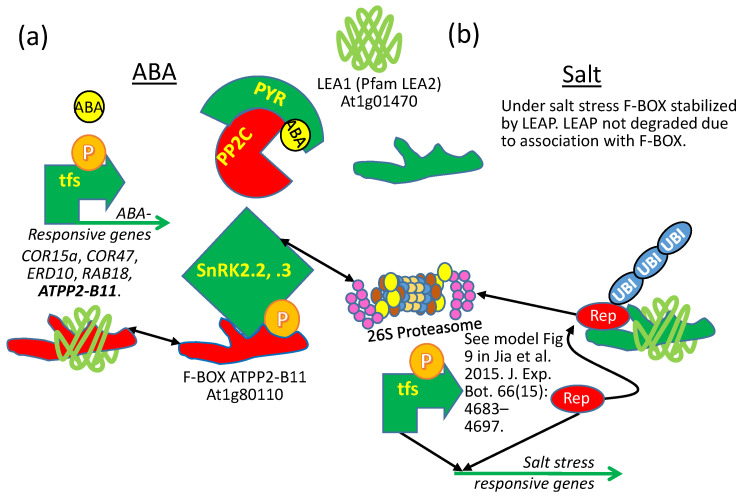
Alteration of phenotypes for a LEAP and one of its client proteins, depending on the physiology of the cell being examined. (**a**) The AtPP2-B11 F-BOX protein targets the SUCROSE NON-FERMENTING KINASEs (SnRK2.2, 2.3), positive regulators of ABA signal transduction, reducing the titer of these kinases in the cell. Both *atpp2-b11* RNAi lines and LEA1-overexpressing lines have enhanced ABA sensitivity. Reducing the F-BOX titer through RNAi stabilizes the SnRK2s titer and results in increased ABA sensitivity. When the LEA1 binds AtPP2-B11, it may sequester the F-BOX protein away from SnRK2.2 and 2.3, preventing their polyubiquitination, and also increasing ABA sensitivity. (**b**) ATPP2-B11 F-BOX overexpression results in greater salt tolerance as does LEA1 overexpression. Reasons for the similar phenotypes under salt stress are, at this time, a matter of speculation, but a model presented in the reference supplied in the figure suggests that a transcriptional repressor is targeted by the F-BOX, and the F-BOX is stabilized during salt stress by LEA1. Green: protein stimulatory for ABA sensitivity or salt tolerance; Red: protein inhibitory for ABA sensitivity or salt tolerance; “P” = phosphorylated; tfs: transcription factors; PP2C: PROTEIN PHOSPHATASE 2C; SnRK: SUCROSE NON-FERMENTING RECEPTOR KINASES; PYR: PYRABACTIN RESISTANT PROTEIN; Rep: transcriptional repressor; UBI: UBIQUITIN.

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
