# Peer review of "Late Embryogenesis Abundant Protein–Client Protein Interactions"

_plants, 2020, doi:10.3390/plants9070814_

Round 1
Reviewer 1 Report
This is a very comprehensive literature review on Late Embryogenesis Abundant proteins. Considering the wealth of studies that have been published over the past years, the authors did an impressive, very thorough search for existing literature. There are very few references missing, and I discovered some in this review that I missed myself. The figures are a useful support,and the supplemental tables are very nice. Undoubtedly, this review will be an important contribution to the field.
I only have some minor remarks:
- Many abbreviations are introduced which makes the reading difficult at times. Some of these might be removed, especially when they are not used often.
- Some awkward sentences exist with many commas: Line 70-72, 147-150
- It would be useful for the reader to have a more detailed explanation of what the liquid-liquid phase separation is.
- I would suggest to remove the exclamation marks.
- Line 221, you can add Boudet et al 2006 (see below)
- Line 311: it is true that LEAPs are intrinsically disordered in the hydrated environment, but several studies show that they change structure when the environment is changed, making binding with its alpha-helices much more likely.
- Line 804: post-transcriptional regulation of LEAP, you can add Verdier et al 2013 (see below).
- References (until 168): some references have titles with capital letters (10, 13, 19, 42, 56, 62, 75, 82, 110, 138, 156). I think that the references from 169 to 384 should be added to the supplemental table rather than the reference list.
If you want to complete the literature list, here are three more papers on LEA proteins:
-Verdier, J et al (2013). A regulatory network-based approach dissects late maturation processes related to the acquisition of desiccation tolerance and longevity of Medicago truncatula seeds. Plant Physiology 163: 757-774. Shows a differential accumulation (>20 days) between LEA transcripts and proteins during seed maturation, suggesting the existence of PTM.
-Chatelain E et al (2012) Temporal profiling of the heat stable proteome during late maturation of Medicago truncatula seeds identifies a restricted subset of late embryogenesis abundant proteins associated with longevity. Plant Cell and Environment, 35: 1440-1455. Shows temporal differences in the accumulation of LEA proteins during seed maturation and identifies those that are most abundant.
-Boudet J et al (2006) Comparative analysis of the heat stable proteome of radicles of Medicago truncatula seeds during germination identifies late embryogenesis abundant proteins associated with desiccation tolerance. Plant Physiology 140: 1418–1436. Shows the change in secondary structure of two seed-specific LEA proteins from unordered-to-ordered structure during fast and slow drying.
Reviewer 2 Report
In my opinion the review proposed will be useful for several researchers who carry out studies on these topics. In particular, I like the choice of the authors to present, in some parts, the review with questions and short answers. The supplementary material provides a great deal of information. However, I suggest to the authors to improve the quality of the figures, several of them are confused and the readers may have difficult to understand them, in particular the Figure 2, Figure 3, Figure 4, Figure 5, Figure 6, Figure 7 and Figure 8.
Reviewer 3 Report
The manuscript is a timely contribution to the field of anhydrobiosis and evidence for the hypothesis of LEAPs regulating the function of client proteins rather than ‘just’ protecting them is compellingly outlined. The authors have performed a thorough review on the current state of knowledge, and lack of it, detailing the potential mechanisms by which LEA proteins confer stress tolerance to mainly plant systems. I enjoyed reading the manuscript and have only a few minor suggestions for improvement. Sometimes I found myself scrolling back and forward to look up a given abbreviation. I think the text may read easier if the authors would use less abbreviations especially the ones that are not utilized too often in the text (e.g. EPP, FBM, CC), but this is merely a suggestion. Furthermore, the authors may want to specify which classification scheme for LEA proteins is being used in the text, since several nomenclatures have been proposed.
Line specific remarks:
49: I think ‘has’ should read ‘have’.
50-52: This sentence was hard for me to understand and might reworded.
90: Do you need ‘is’ at the end of the sentence?
139: You may want to mention that several anhydrobiotic animals (e.g. certain tardigrades, rotifers) appear to lack trehalose and seek to use IDPs and other proteins exclusively for protection. However, the physiochemical properties of the used IDP might differ for species that utilize trehalose from those found in animals that lack the sugar.
173: I wonder what protein fraction of the total proteome is sensitive to desiccation and expect that similar to the ‘heat-stable fraction’ a lot of proteins if dried under crowed conditions may actually experience only minor damages during desiccation and rehydration.
191: I would keep in mind that LEAPs need to exhibit certain physiochemical properties to be able to undergo LLPS such as being relatively hydrophobic compared to most LEA proteins. In other words, I expect only a small subset of LEA proteins to participate in this phenomenon.
703: Some word is missing between ‘ID’ and ‘LEA’.
Reviewer 4 Report
The review is very well written, nicely summarizes important findings in the field, and critically outlines some of the “conflicting” results in the field. For my taste, some paragraphs are a bit overstated and vague (for example first paragraph on page 4) but this might be conceived differently by other scientists. Overall, it is a very good and well written review. I have some additional comments that could be addressed before publication:
Line 67 “without it?” does not make sense for molecules in my opinion. Small molecules cannot “have” an EPP. The header also does not really represent what is written in the paragraph below, which is not about organisms but mainly about fragile biological materials (FBM), so larger biomolecules. I suggest changing the header to something that is more clear.
Line123: You cannot crystalize intrinsically disordered proteins that per se do not have (tertiary) structure (i.e. it is not difficult but impossible)
Line 205: Why should LEAP bind the client proteins with their ID region and not through a partially folded domain (if there are any)?
Line 211: It does not really make sense that in stressful conditions, the client proteins would just passively diffuse into preformed phase separated compartments. Would that not be way too slow to actually have a protective function on the client protein? Are the client proteins not actively concentrated and encapsulated through LLPS?
Line378: There is “.” missing at the end of the sentence or the formatting of the sentence is wrong (i.e. no new paragraph here).
Line 532: I think the text in Figure 8 is too big. It is a bit confusing to understand which text belongs to which sub panel (especially between a and b).
Line 854: Part of the header is in bold text
